**Subject Category:**
Biology (whole organism)

ecology

stereo-video measurements, visual estimation, diver-operated video systems, megafauna, top predator

**Author for correspondence:**
Charlie Huveneers
e-mail: charlie.huveneers@flinders.edu.au

# Eyes on the size: accuracy of visual length estimates of white sharks, *Carcharodon carcharias*

Cameron May, Lauren Meyer, Sasha Whitmarsh and Charlie Huveneers

Southern Shark Ecology Group, College of Science and Engineering, Flinders University, GPO Box 2100, Adelaide, South Australia 5001, Australia

CM, 0000-0001-5399-8079; LM, 0000-0003-0374-9941; SW, 0000-0001-8934-2354; CH, 0000-0001-8937-1358

Visual estimates have been used extensively to determine the length of large organisms that are logistically challenging to measure. However, there has been little effort to quantify the accuracy or validity of this technique despite inaccurate size estimates leading to incorrect population assessments and misinformed management strategies. Here, we compared visually estimated total length measurements of white sharks, *Carcharodon carcharias*, during cage-diving operations with measurements obtained from stereo-video cameras and assessed the accuracy of those estimates in relation to suspected biases (shark size, and observer experience and gender) using generalized linear mixed-models and linear regressions. Observer experience on board cage-diving vessels had the greatest effect on the accuracy of visual length estimates, with *scientists* being more accurate (mean accuracy ± standard error: 23.0 ± 16.5 cm) than *crew* (39.9 ± 33.8 cm) and *passengers* (49.4 ± 38.5 cm). Observer gender and shark size had no impact on the overall accuracy of visual length estimates, but *passengers* overestimated sharks less than 3 m and underestimated sharks greater than 3 m. Our findings show that experience measuring animals is the most substantial driver of accurate visual length estimates regardless of the amount of exposure to the species being measured. *Scientists* were most accurate, even though *crew* observe white sharks more frequently. Our results show that visual length estimates are not impacted by shark size and are a valid measurement tool for many aspects of *C. carcharias* research, provided they come from people who have previously been involved in measuring animals, i.e. *scientists*.

# 1. Introduction

Morphometry has extensive applications ranging from taxonomy, where species shape and form are used to identify and classify species [1,2], to biology and ecology, where an individual's size is necessary to assess growth rate, size–weight relationships, size- and age-at-maturity and ontogenetic changes in habitat and diet [3,4]. However, due to the wide diversity of species sizes, habitats and behaviours, a variety of methods has been developed to measure organisms, which range from direct measurements (i.e. physically measuring dimensions of an individual) and visual estimates, to more technologically advanced photography-aided methods.

The most commonly used method of obtaining lengths is through direct measurements, as it is highly accurate and applicable across taxa [5,6]. However, direct measurements are not always logistically feasible. Measuring species in the marine environment has additional challenges compared to terrestrial ecology because it can be difficult to safely catch, restrain and measure large aquatic organisms (e.g. cetaceans, large sharks), all of which can have detrimental implications for species with high post-release mortality [7]. Visual length estimates are less invasive, overcoming some of the practical challenges of sampling in the marine environment while also providing an ethical refinement to obtaining length measurements [8]. However, many studies using visual length estimates to assess large megafauna do not quantify the accuracy of the estimates (e.g. [9–11]). When accuracy is quantified, visual length estimates have been shown to have considerable variance, even between experienced observers [12–14]. A previous study has found that visual length estimates of whale sharks, *Rhincodon typus*, were underestimating the true size of the organism by several metres, which may have led to an underestimation of the number of mature *R. typus* in Ningaloo Reef, in Western Australia [14]. This underestimation may have distorted previous population models by wrongly inferring population growth due to underestimation of length and size-at-maturity. Accurate population modelling for endangered species such as *R. typus* is vital to assess population status and develop effective conservation strategies [15]. Thus, unquantified errors in visual length estimates may have broad implications and studies need to assess the accuracy and skew of visual length estimates and identify factors that might influence this accuracy.

Several factors may affect visual length estimates, including the size of the animal. The lengths of small species (less than 50 cm) are typically easier to estimate than large species, e.g. error margins of 3.1% and less than 1 cm for species 8–35 cm [16,17]. Divers are also capable of correctly categorizing target species into 10 cm size classes [18]. By contrast, *R. typus* size estimates were often inaccurate and changed from overestimating small sharks (less than 5 m) to underestimating large sharks (greater than 7 m) with the most accurate estimates produced for sharks of intermediate length (i.e. 4–6 m) [14,15].

Observer experience with the organism can also affect the accuracy of visual length estimates, with accuracy typically improving with training in estimating sizes and exposure to the organism [18,19]. This is especially relevant for studies using non-specialist volunteers, i.e. citizen science, because of the often-criticized reliability and accuracy of the data [20,21]. Inaccurate length estimates are not limited to inexperienced observers, as experienced observers can also have considerable error in their estimates [19,22] and make errors of more than 1 m in large species [14,15]. Therefore, studies using visual length estimates must not simply rely on using experienced observers, but should assess the accuracy of those estimates and quantify how much training or experience might be necessary to ensure accurate estimates.

Gender differences in spatial awareness tasks are considered to be one of the largest differences in all cognitive abilities between males and females [23,24], with the size of this effect changing depending on the type of spatial skill measured [25]. Differences in the ability to estimate distances and lengths have yielded varying results. Most studies suggest that males are more accurate at estimating distance than females [26–28], but the opposite has also been shown [29]. These studies showcase that gender can influence distance estimation, yet how this factor influences the ability to visually estimate the length of an organism has yet to be explored. With both men and women providing length estimates in research, understanding if, and how inherent gender differences influence visual length estimates is vital when considering the accuracy of such estimates.

The white shark, *Carcharodon carcharias*, is a species for which visual length estimates are routinely used, including in assessments of residency and habitat use [30,31], fine-scale position and activity [32,33], population size and survival rates [34,35], and population dynamics and ontogenetic shifts [11,36–38]. Due to the difficulties of catching, restraining and measuring individuals, and the safety risks associated with handling large, potentially dangerous animals [8,39,40], visual estimation

techniques are favoured over direct measurements. Previous research has attempted to assess the accuracy of such estimates for *C. carcharias* [41], but was limited to a small sample size of three observers and inaccuracy by an average of 42 cm. Further research on the ability to accurately estimate *C. carcharias* length and the factors that impact such accuracy is required, especially since length is a critical component in studies related to the conservation status of *C. carcharias*. For example, the size of the Australian *C. carcharias* population has recently been estimated using close kin mark−recapture that requires information on age and maturity [42,43]. As both of these variables are informed by length, inaccurate length estimates when collecting the genetic samples used in the study could have led to an erroneous population size estimate.

Stereo-photogrammetry has recently been used to measure a broad range of shark species including *C. carcharias* [44], *R. typus* [14], oceanic whitetip sharks, *Carcharhinus longimanus* [8] and sandbar sharks, *Carcharhinus plumbeus* [45], as it replaces the need to capture individuals. Stereo-photogrammetry uses two cameras recording simultaneous footage and specialized software to record point-to-point measurements [46]. The method is effective underwater, with Harvey *et al.* [47] quantifying the accuracy of the technique by comparing stereo-obtained measurements of southern bluefin tuna, *Thunnus maccoyii*, to true length measured directly, and found a difference of 0.16%. Despite the proven ability of stereo-camera technology to capture accurate measurements of *C. carcharias*, visual estimates are still favoured as a method due to the ease and reduced cost.

Obtaining the accurate length of *C. carcharias* is vital to understand key life-history parameters and population dynamics, yet the accuracy of visually estimating *C. carcharias* length has been poorly investigated. This study determined observer ability to visually estimate *C. carcharias* total length (TL), and assess how shark length, and observer experience and gender affected the accuracy and skew of these estimates.

# 2. Methods

## 2.1. Study site

Stereo-camera footage and visual length estimates of *C. carcharias* were collected at the Neptune Islands Group Marine Park (35°149 S; 136°049 E), South Australia from February 2017 to May 2018. The Neptune Islands are a key aggregation site for *C. carcharias*, located approximately 60 km south of Port Lincoln at the mouth of Spencer Gulf. The consistent year-round shark population supports Australia's only *C. carcharias* cage-diving industry, consisting of three operators, with approximately 10 000 tourists partaking in the experience annually [48].

## 2.2. Stereo-camera measurements

Measurements of *C. carcharias* were obtained with stereo-camera technology, consisting of two GoPro Hero4 Silver video cameras with long-life battery backpacks in custom-made SeaGIS housings and mounts (www.seagis.com.au). The cameras were set 76 cm apart along a metal bar and angled 8° inward, with each camera's field of view set at 130°. Sharks were filmed with the stereo-camera during cage-diving operations, from inside the shark-diving cage, or held underwater from the surface using a pole attachment. Footage collected allowed for image analysis in the specialized EventMeasure software (www.seagis.com.au), which allows for accurate point-to-point measurements [49,50].

The stereo-camera was calibrated with the CAL-Stereo Camera Calibration software, a distance bar and calibration cube (1000 × 1000 × 500 mm; www.seagis.com.au). With the stereo-camera immobilized underwater, the calibration cube was manually rotated in different positions within the field of view of both cameras to ensure a range of calibration points at varying angles and distances. The distance bar is a horizontal bar 1.3 m long that is used to independently check calibration accuracy by constantly rotating and shifting the angle of the bar while moving closer to the camera. Recalibration of the stereo-camera was undertaken three times during the study period, with approximately six months between each calibration, to avoid inaccurate measurements due to shifts in camera angles from handling and operating the stereo-camera units. The mean level of error between true length and stereo-camera measurement was 0.42 cm, indicating a very high level of accuracy and reliability.

Footage of individual sharks was assessed for its suitability based on the orientation of *C. carcharias* in relation to the stereo-camera, visibility and the clarity of the points used for measurement (snout and caudal fin). For each suitable pass (i.e. when a shark was fully extended, with all measurement points visible in the frame), the shark was identified using natural markings including amputations, the shape of the trailing edge of the dorsal fin, markings on the flank, upper and lower caudal, and differences in the countershading boundary [51]. Total length was measured for each pass from the shark's snout to the intersection of the line from the snout to the caudal peduncle and the line joining the tips of the lower and upper caudal fin [44]. Each individual shark was measured in as many suitable passes as possible during each video, which resulted in a different number of measurements taken for each shark. Average measurement of all passes was calculated for each shark.

## 2.3. Visual length estimates

Observers were surveyed in accordance with ethics approval provided by Flinders University Social and Behavioural Research Ethics Committee (approval number: 7862). Field sampling approval was granted by the South Australian Department for Environment and Water (permit number: Q26612-2).

Observers were categorized by their experience and role on cage-diving vessels (*passenger*, *crew*, *scientist*), and gender (*male* or *female*). *Passengers* were paying customers partaking in cage-diving and who had seen *C. carcharias* on less than three occasions. *Crew* were staff members who have worked for a cage-diving operator for at least six months and who have frequently interacted with *C. carcharias*. *Scientists* were authors of peer-reviewed papers on *C. carcharias* or other shark species and who have previous experience measuring sharks in the field. Observers were instructed to provide a visual length estimate from the tip of a shark's snout to the upper caudal fin.

## 2.4. Statistical analysis

Statistical analyses were conducted using IBM SPSS Statistics and R studio using the lme4 R package [52], with significance values set at $p < 0.05$.

Generalized linear mixed-effects models (GLMM) were used to determine if the accuracy of visual estimates, measured as the absolute difference between visual estimates and stereo-measurements (ABSdiff), varied significantly with *C. carcharias* length (covariate), and observer experience and gender (fixed factors). Since accuracy could have changed with the number of sharks sighted (as people's ability to estimate shark length might improve after repeated exposure), the number of sharks sighted per day was included as a fixed factor. Shark ID was also included as a random effect.

The data were scaled and centred prior to the analyses in R to eliminate biases and normalize the scales in the variables. The error structure of a GLMM corrects for the non-independence of statistical units due to some *C. carcharias* having their length estimated by several observers, and permits the random-effects variance explained at different levels of clustering to be decomposed. We determined the most appropriate statistical family and error distribution (Gaussian with log link) by examining the distribution of the response variable and visually inspecting the residuals for the saturated models. We ran all models for all possible combinations of factors, and compared their relative probability using Akaike's information criterion corrected for small sample size (AIC$_c$) [53].

Linear regressions on the non-transformed data were used to assess the relationship between accuracy (difference between visual length estimates and stereo-measurements) and shark length. Regressions were applied to all estimates combined and from each experience level.

## 3. Results

Stereo-footage of 35 *C. carcharias* was collected between February 2017 and May 2018. Sharks were between 2.41 and 4.70 m TL (mean $\pm$ standard error: $3.42 \pm 0.63$ m; figure 2), with most (41%) between 3.00 and 3.50 m TL (figure 1). A total of 124 observers (54 females and 70 males) provided 322 visual length estimates (ranging from 1.70 to 5.50 m TL; figure 2). Most observers were classified as *passengers* (87%) who consequently provided the majority of estimates (66%), followed by *crew* (23%) and *scientists* (11%), with *crew* and *scientists* estimating 24 sharks (approx. 70% of sharks observed).

Observers' experience was the only factor influencing measurement accuracy ($w$AIC$_c$ = 0.47; tables 1 and 2), and explained approximately 5.9% of the variance in the data. *Scientists* were the most accurate (absolute difference: $23 \pm 16.5$ cm), followed by *crew* ($39.9 \pm 33.8$ cm) and *passengers* ($49.4 \pm 38.5$ cm;

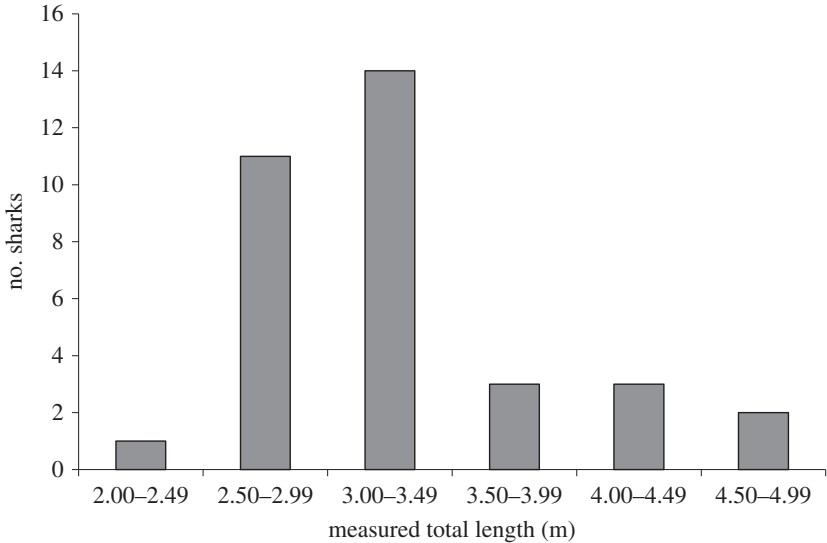

**Figure 1.** Frequency histogram of *C. carcharias* measured in the study.

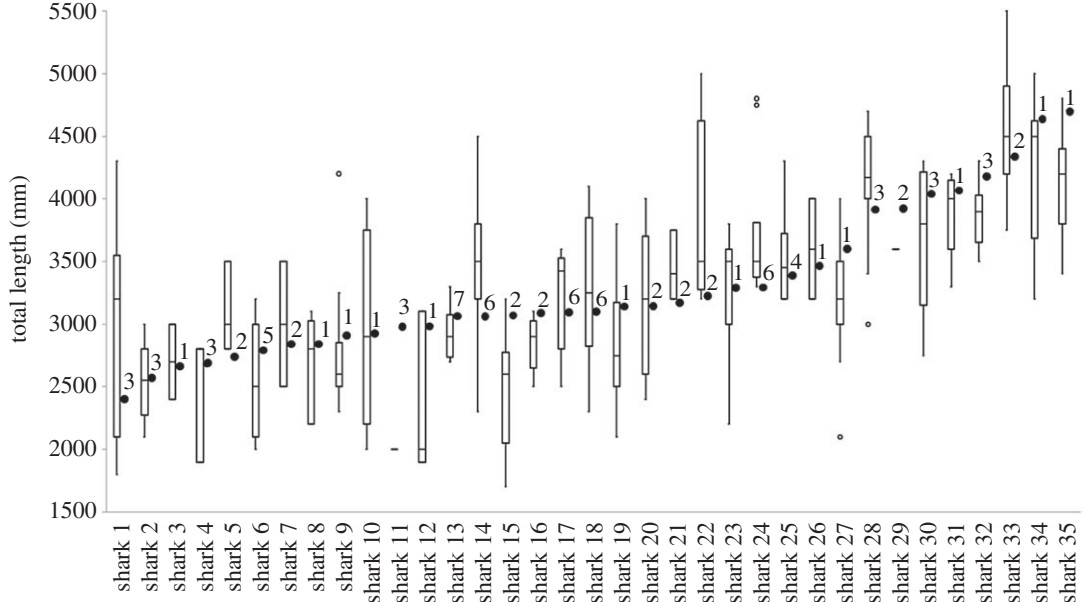

**Figure 2.** Visual size estimates and stereo-video measurements collected of *C. carcharias*. Boxplots represent the range of estimates given from all participants (not distinguished between *passenger*, *crew* and *scientists*) for each *C. carcharias* (listed 1–35). The median values are indicated by the bold horizontal bar; the length of the box is the inter-quartile range; whiskers represent quartiles; and white circles are extreme values. Black circle represents the accompanying mean stereo-video measurement for each *C. carcharias*, with superscript numbers representing the number of stereo measurements. Sharks are ordered from smallest to largest based on the stereo-video measurement.

figure 3*a*). Gender, shark length and number of sharks per day were not included in the top-ranked model, indicating they were unlikely to strongly affect the accuracy of size estimates (figure 3).

The linear regressions showed that *passengers* overestimated sharks less than 3 m and underestimated sharks greater than 3 m (figure 4); however, this trend was very weak ($R^2 = 0.06$) and was less apparent with *crew* and *scientists* ($R^2 = 0.02$, and 0.002, respectively).

## 4. Discussion

The accuracy of visual length estimates and the factors that influence it has seldom been assessed. Our study revealed that observer experience had the greatest impact on the accuracy of visual length estimates of *C. carcharias*, with *scientists* providing the most accurate length estimates (within approx. 20 cm of

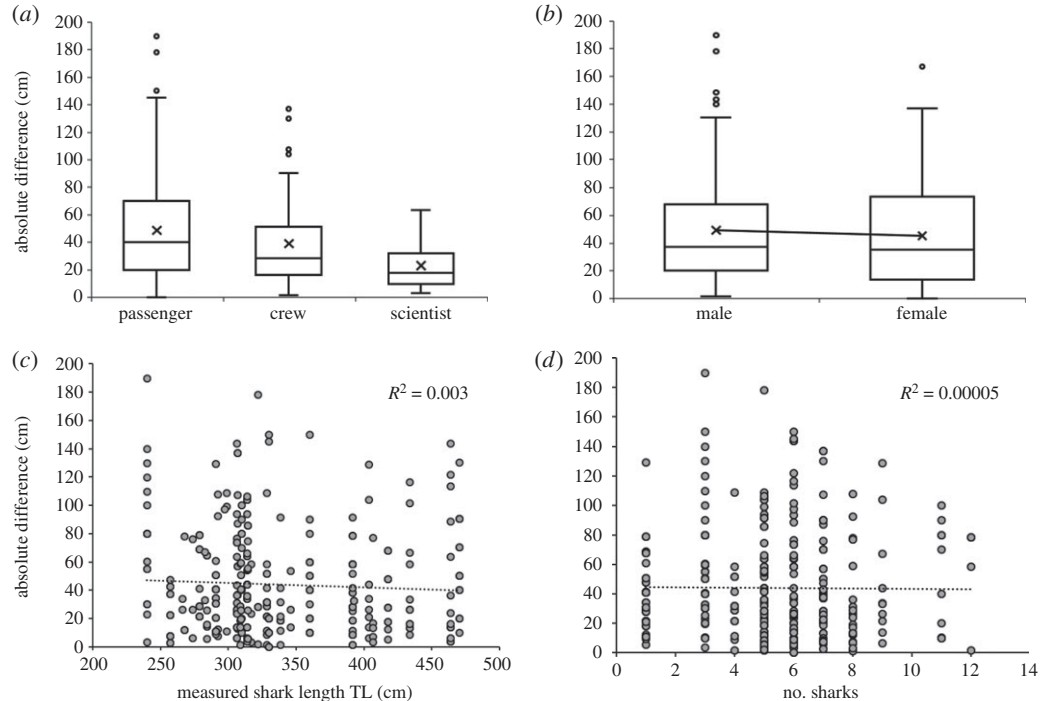

**Figure 3.** Absolute difference between visual estimate and measured length against (*a*) observer experience; (*b*) observer gender; (*c*) shark TL; and (*d*) number of sharks sighted per day. Boxplots include median values, indicated by the bold horizontal bar; the length of the box is the inter-quartile range; whiskers represent quartiles; and white circles are extreme values. In (*c*) and (*d*), dotted line represents linear regression with $R^2$ value shown on the top right corner.

**Table 1.** Top five ranked linear mixed-effects models of factors influencing the absolute difference between the lengths estimated visually and by stereo-video measurements. *k*, number of model parameters; $AIC_c$, Akaike's information criterion corrected for small sample size; $\Delta AIC_c$, difference in $AIC_c$ between the current and the top-ranked model; $wAIC_c$, model probability; $R_m$, marginal (fixed effects) $R^2$; $R_c$ conditional (random effect) $R^2$. All models include shark ID as a random effect. Best performing model is highlighted in italics. Complete model set can be found in electronic supplementary material, table S1.

| | *k* | $AIC_c$ | $\Delta AIC_c$ | $wAIC_c$ | $R_m$ | $R_c$ |
|---|---|---|---|---|---|---|
| *absolute difference ~ experience* | *5* | *897.20* | *0* | *0.47* | *5.87* | *13.52* |
| absolute difference ~ shark length + experience | 6 | 898.59 | 1.73 | 0.19 | 5.99 | 13.94 |
| absolute difference ~ experience + sharks per day | 6 | 899.12 | 1.99 | 0.17 | 5.86 | 13.94 |
| absolute difference ~ shark length + experience + sharks per day | 7 | 900.90 | 3.70 | 0.07 | 5.99 | 14.38 |
| absolute difference ~ gender + experience | 8 | 902.00 | 4.80 | 0.04 | 6.28 | 13.86 |

**Table 2.** Estimated coefficients (*β*) and their standard errors (s.e.), *t*-values of factors included in the top-ranked model.

| fixed effects | *β* | s.e. | *t*-value |
|---|---|---|---|
| intercept | −0.150 | 0.122 | −1.228 |
| scientist | −0.471 | 0.197 | −2.390 |
| passenger | 0.292 | 0.127 | 2.290 |
| crew | 0 | | |
| random effect | variance | s.d. | |
| shark ID | 0.077 | 0.278 | |
| residual | 0.875 | 0.936 | |

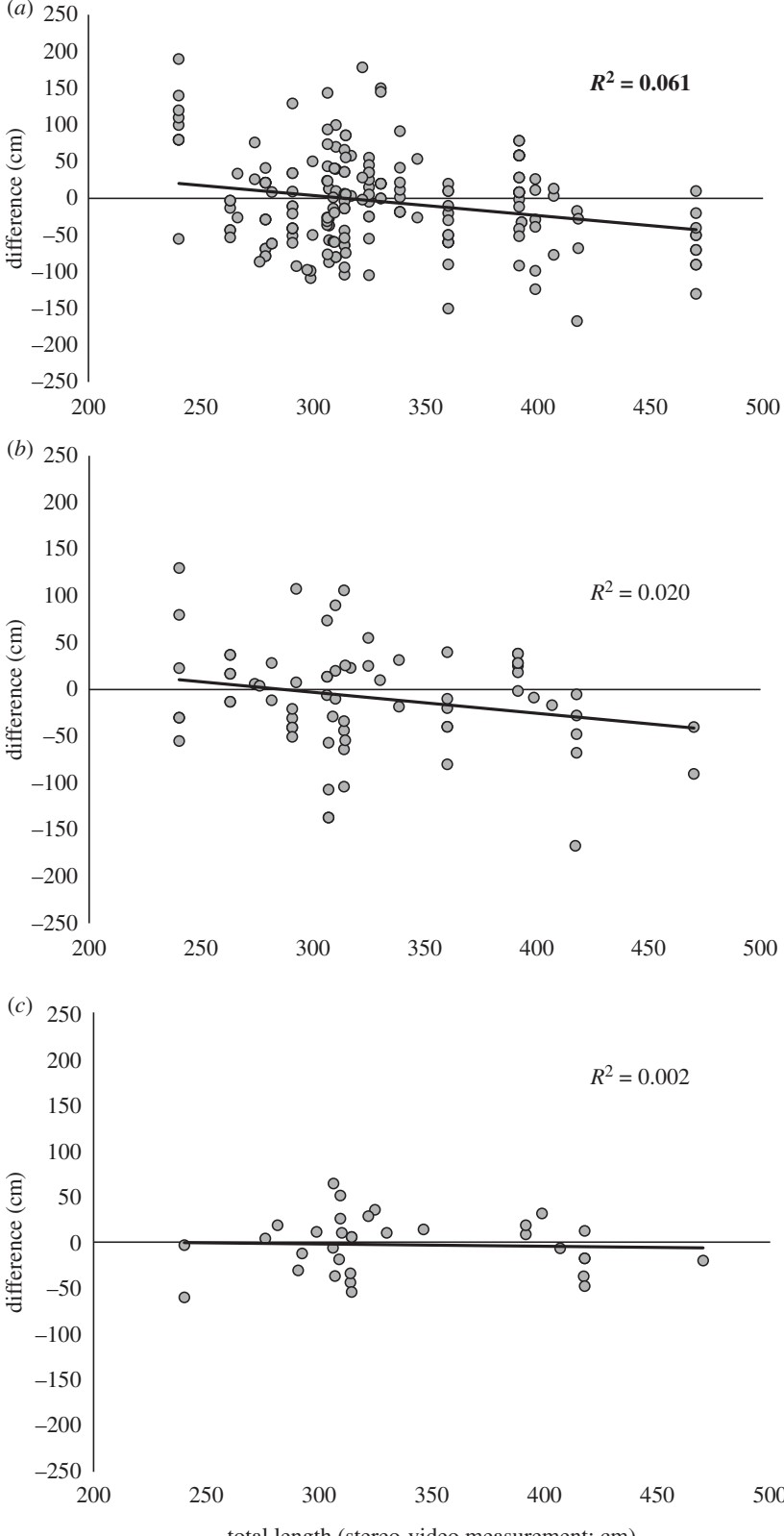

**Figure 4.** Relationship between measured total length and the difference between visual estimate and stereo-video measurement. (*a*) *Passenger*, (*b*) *crew* and (*c*) *scientists*. Black line shows line of best fit, $R^2$ in bold indicates significant linear regression ($p < 0.05$).

stereo-measurements) followed by *crew* (within approx. 40 cm) and *passengers* (within approx. 50 cm). The size of *C. carcharias* and observer gender had no effect. Our findings show that visual length estimates can be used in research that is not sensitive to biases of 20 cm when estimates are made by

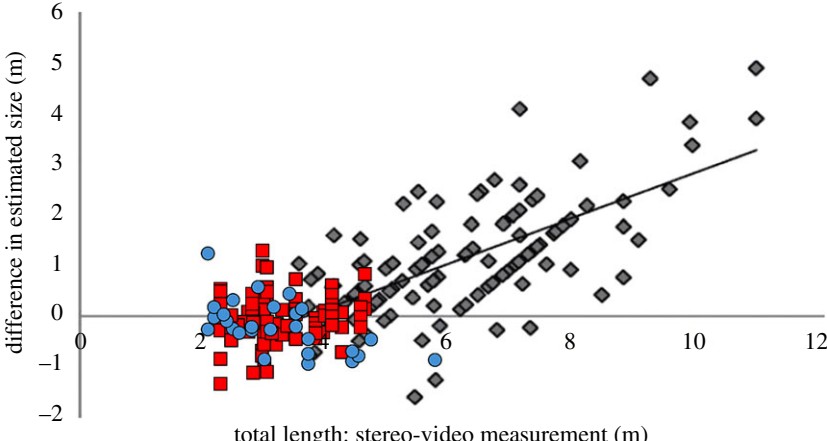

**Figure 5.** Comparison of the relationship between measured TL and the difference between visual estimate and stereo-video measurement across the present study (red squares), [41] (blue circles) and [14] (grey diamonds). Data from the present study only include *crew* and *scientist* to make it comparable to Leurs *et al.* [41] and Sequeira *et al.* [14], which did not include citizen science estimates.

scientists, or 50 cm when based on citizen science, such as studies categorizing sharks into life stage categories (e.g. neonate, small juvenile, large juvenile, adult; [11,37]). Visual length estimates can also be useful at detecting growth across multiple years, e.g. growth of *C. carcharias* that are regularly resighted at aggregation sites (e.g. [54]).

Despite *crew* spending the most time around *C. carcharias* and being exposed to *C. carcharias* on a day-to-day basis, their accuracy was closest to *passengers* with mean error twice as large as that of *scientists*. Low accuracy from *crew* was somewhat surprising but is probably due to *crew* being required to focus on a number of non-shark related tasks during cage-diving operations and therefore being less experienced at scientifically estimating shark size. By contrast, *scientist* experience measuring sharks during other studies and attention to details from the standard rigour required during scientific investigations were sufficient to improve the accuracy of visual estimates. *Scientists'* accuracy was better than *crew's* even though *scientists* were less often exposed to *C. carcharias* than *crew* and had never been able to validate estimates against measurements of *C. carcharias* or other species greater than 2.5 m.

*Crew's* accuracy in the present study (39.9 $\pm$ 33.8 cm) was similar to that of crew from another cage-diving vessel in South Africa (42.0 $\pm$ 32.0 cm) [41]. Although Leurs *et al.* [41] had a much lower sample size than our study (3 observers and 27 estimates versus 124 participants and 304 estimates, respectively), the similarity in results suggests that the accuracy of *C. carcharias* visual size estimates from crew members of a shark cage-diving boat is approximately 40 cm. As *passengers* were unsurprisingly the least accurate (49.4 $\pm$ 38.5 cm), data relying on citizen science and non-specialist volunteers to estimate size of *C. carcharias* should be used cautiously. Errors of approximately 50 cm in *C. carcharias* research could result in under or overestimation of mature individuals, which can impact population assessments, as suggested with *R. typus* [14].

The accuracy of *passenger* estimates did not improve with short-term experience, measured as the number of *C. carcharias* observed. Although our findings indicate that experience visually estimating sharks is necessary to provide accurate visual length estimates, *passengers* did not improve after seeing or in the presence of more sharks, or when estimating multiple sharks. Training and calibration has been shown to improve the accuracy of volunteers and non-specialist divers [20]; however, this training is dependent on the conscious calibration and refinement of divers' estimates, which was intentionally not provided in this study (i.e. passengers were never told the size of the sharks). As most sharks sighted at the Neptune Islands are typically between 3 and 4 m (approx. 70%; [11]), participants were unlikely to have had the opportunity to estimate sharks from a broad range of sizes, which is an equally important aspect of calibration. The calibration required to improve non-trained participants involves observing objects of varying lengths (e.g. 2–6 m), multiple times before being able to produce accurate length estimates [20]. As passengers in this study did not observe multiple sharks from a broad range of lengths in multiple instances, they did not receive the necessary calibration to improve their estimation ability.

Overall, the size of *C. carcharias* did not affect the accuracy of observer estimates. The accuracy in our study was similar to that of 3–4 m *R. typus* ([14], figure 5), which is the same size class as most *C. carcharias* observed in the present study. However, accuracy in *R. typus* decreased in sharks greater than 6 m, suggesting that it may be harder to accurately estimate the size of large organisms

(e.g. whale sharks greater than 8 m) than 3–5 m individuals. This could also be related to visual perception and the role of a mental image [55]. It might be easier for observers to estimate the size of a 3–5 m organism, as they may have mental images of objects of similar sizes (e.g. vehicles, furniture) [55]. This is supported by previous research showing that visual size estimates of organisms less than 50 cm can be accurately estimated with little variance (approx. 3.5%, less than 1 cm) [16,17] versus large organisms (greater than 8 m) that can have large variance (approx. 40%, 4–5 m) [14,15]. While the magnitude of the inaccuracy does not increase with *C. carcharias* size, *passengers* overestimated small *C. carcharias* (less than 3 m) and underestimated large *C. carcharias* (greater than 3 m), as seen in previous studies [14,15]. The same trend has now been reported in potentially dangerous species (i.e. *C. carcharias*) [41] and non-threatening species (*R. typus*) [14,41,56], indicating that public perception of a species does not impact how they are visually estimated. The opposite was hypothesized due to the charismatic nature, public perception and negative portrayal of *C. carcharias* in the media [56–58]. It is also possible that *passengers* in the present study did not have a negative perception of *C. carcharias*, as shark-diving tourists are typically informed and passionate about sharks, looking to expand their knowledge and extend their awareness of the species [59]. *Crew* and *scientists* did not show bias in their estimates depending on the size of the sharks, indicating that length estimates from experienced observers can be accurate, regardless of the size of *C. carcharias*. This finding differs from *R. typus* and indicates that visual length estimates of *C. carcharias* are more valid than estimates of larger species.

The effect of gender on accuracy could not be investigated for each experience level due to the small number of female *crew* (6) and *scientist* (1) observers. However, when combined across groups, observer gender did not significantly affect the accuracy of visual length estimates, despite previous studies establishing clear gender differences in spatial awareness and length estimation tasks, and recognition of psychological differences between genders [26–28]. The lack of gender bias observed in the present study might be explained by studies showing that although males and females use different methods to perform cognitive tasks [60], both genders have similar cognitive abilities [61].

## 5. Conclusion

Our study reveals that people's ability to accurately estimate *C. carcharias* length is not influenced by *C. carcharias* size or observer gender, but is affected by observer experience in estimating lengths. *Scientists*' size estimates were on average within approximately 20 cm of the stereo-measured length, while *crew* and *passengers* were within approximately 40 and 50 cm, respectively. These findings can be applied to studies that use visual length estimates of *C. carcharias* to assess if the accuracy of length estimates is sufficient for the aims of the intended study. Future research should increase the number of estimates from *scientists* to further investigate the effect of confidence using a group that can objectively reflect on their ability to estimate shark size.

Ethics. Ethics approval provided by Flinders University Social and Behavioural Research Ethics Committee (approval no. 7862). Animal ethics approval provided by Flinders University Animal Ethics Committee (approval no. E435/16). Field sampling approval by the Department of Environment and Water (permit no. Q26612-2).

Data accessibility. Visual length estimate and stereo-camera data are available in the electronic supplementary material 'CM_Data'.

Authors' contributions. C.M. designed the study, collected and analysed the data, produced the tables and figures, and wrote the manuscript. C.H. contributed to the experimental design, collected data and reviewed drafts of the manuscript. L.M. and S.W. contributed to the experimental design, collected data and reviewed drafts of the manuscript.

Competing interests. All authors have no competing interests.

Funding. Funding was provided by DEW and supported by the white shark cage-diving industry.

Acknowledgements. Staff and crew from Rodney Fox Shark Expeditions and Calypso Star Charters are thanked for logistical support during fieldwork, as are field volunteers and observers.

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
