## [Reviewer comments · Royal Society Open Science]

Review History

RSOS-190456.R0 (Original submission)

Review form: Reviewer 1 (John Musick)

Is the manuscript scientifically sound in its present form?

Yes

Are the interpretations and conclusions justified by the results?

Yes

Is the language acceptable?

Yes

Is it clear how to access all supporting data?

Yes

Do you have any ethical concerns with this paper?

No

Have you any concerns about statistical analyses in this paper?

No

Recommendation?

Accept with minor revision (please list in comments)

Comments to the Author(s)

Interesting and needed paper. Good job! References Section needs some cleaning up. Some incomplete, at least one has date listed at end instead of after authors.

Review form: Reviewer 2

Is the manuscript scientifically sound in its present form?

Yes

Are the interpretations and conclusions justified by the results?

Yes

Is the language acceptable?

Yes

Is it clear how to access all supporting data?

Yes

Do you have any ethical concerns with this paper?

No

Have you any concerns about statistical analyses in this paper?

No

Recommendation?

Accept with minor revision (please list in comments)

Comments to the Author(s)

General Comments: This is an interesting manuscript that compares visual estimates of white sharks by three groups (scientists, crew and passengers) and compares with measurements taken by stereo-camera system. The manuscript is well written and the results and discussion are suitable. More information is required in the methods section regarding validity/accuracy of the stereo-camera system (see below).

Specific Comments:

P5 - L22-55. Regarding accurate measuring of white sharks, it has been previously demonstrated that using stereo-video is a suitable and accurate method however there is no mention of this. There is no mention of other studies that have employed non-visual methods for estimating white shark size, even though there are published studies that document direct measurements rather than visual, there should be at least a sentence in the Intro discussing these other studies of direct measurement. See for example: Harasti, D., Lee, KA., Laird, R., Bradford, R. and Bruce., B. (2016). Use of baited remote underwater video systems (BRUVs) to estimate relative abundance and size of juvenile white sharks *Carcharodon carcharias*. *Marine and Freshwater Research*. <http://dx.doi.org/10.1071/MF16184>.

P6 37-55: There needs to be much more information provide about the stereo-camera system used

in regards how were they calibrated, what were the calibration tests and what was the camera accuracy. Below are some specific questions

How were the cameras calibrated? Was a calibration cube used?

How often was the camera calibration checked? Was it done just once or throughout the entire study? Did the camera need to be recalibrated at all?

Was a known length scale bar used to test calibration?

How do you know that the cameras were providing accurate estimates?

Provide error estimates for the calibration tests and provide some data summarising the calibration tests. This will help demonstrate the accuracy of measurements from the stereo-camera system.

Can you provide an estimate of the size of the field of view or detection range of the cameras?

How many measurements of each shark were used? Was an average measurement taken based on different passes/angle/frames? Or just one single measurement taken? More info is required.

P7 L 39: members

P10 L line 45: non specialist volunteers to estimate size of *C. carcharias* should be used

Decision letter (RSOS-190456.R0)

08-Apr-2019

Dear Mr May

On behalf of the Editors, I am pleased to inform you that your Manuscript RSOS-190456 entitled "Eyes on the size: Accuracy of visual length estimates of white sharks, *Carcharodon carcharias*" has been accepted for publication in Royal Society Open Science subject to minor revision in accordance with the referee suggestions. Please find the referees' comments at the end of this email.

The reviewers and handling editors have recommended publication, but also suggest some minor revisions to your manuscript. Therefore, I invite you to respond to the comments and revise your manuscript.

- Ethics statement

- Data accessibility

If you wish to submit your supporting data or code to Dryad (<http://datadryad.org/>), or modify your current submission to dryad, please use the following link:
<http://datadryad.org/submit?journalID=RSOS&manu=RSOS-190456>

- **Competing interests**

- **Authors' contributions**

- **Acknowledgements**

- **Funding statement**

Because the schedule for publication is very tight, it is a condition of publication that you submit the revised version of your manuscript before 17-Apr-2019. Please note that the revision deadline will expire at 00.00am on this date. If you do not think you will be able to meet this date please let me know immediately.

When submitting your revised manuscript, you will be able to respond to the comments made by the referees and upload a file "Response to Referees" in "Section 6 - File Upload". You can use this to document any changes you make to the original manuscript. In order to expedite the

processing of the revised manuscript, please be as specific as possible in your response to the referees. We strongly recommend uploading two versions of your revised manuscript:

on behalf of Professor Kevin Padian (Subject Editor)
 openscience@royalsociety.org

Associate Editor Comments to Author:

Two reviewers have commented on your manuscript, and both find merit in publishing the paper; however, referee 2 in particular has a number of comments that you should address in your response to reviewers (and include as modifications in your revised manuscript). Please make the changes clear in your revision, which will help the Editors determine whether additional review is required, though given the relatively minor changes suggested further review appears improbable. We look forward to receiving the revised manuscript.

Reviewer comments to Author:

Reviewer: 1

Comments to the Author(s)

Interesting and needed paper. Good job! References Section needs some cleaning up. Some incomplete, at least one has date listed at end instead of after authors.

Reviewer: 2

Comments to the Author(s)

General Comments: This is an interesting manuscript that compares visual estimates of white sharks by three groups (scientists, crew and passengers) and compares with measurements taken by stereo-camera system. The manuscript is well written and the results and discussion are suitable. More information is required in the methods section regarding validity/accuracy of the stereo-camera system (see below).

Specific Comments:

P5 - L22-55. Regarding accurate measuring of white sharks, it has been previously demonstrated that using stereo-video is a suitable and accurate method however there is no mention of this. There is no mention of other studies that have employed non-visual methods for estimating white shark size, even though there are published studies that document direct measurements rather than visual, there should be at least a sentence in the Intro discussing these other studies of direct measurement. See for example: Harasti, D., Lee, KA., Laird, R., Bradford, R. and Bruce., B. (2016). Use of baited remote underwater video systems (BRUVs) to estimate relative abundance and size of juvenile white sharks *Carcharodon carcharias*. *Marine and Freshwater Research*. <http://dx.doi.org/10.1071/MF16184>.

P6 37-55: There needs to be much more information provide about the stereo-camera system used in regards how were they calibrated, what were the calibration tests and what was the camera accuracy. Below are some specific questions

How were the cameras calibrated? Was a calibration cube used?

How often was the camera calibration checked? Was it done just once or throughout the entire study? Did the camera need to recalibrated at all?

Was a known length scale bar used to test calibration?

How do you know that the cameras were providing accurate estimates?

Provide error estimates for the calibration tests and provide some data summarising the calibration tests. This will help demonstrate the accuracy of measurements from the stereo-camera system.

Can you provide an estimate of the size of the field of view or detection range of the cameras?

How many measurements of each shark were used? Was an average measurement taken based on different passes/angle/frames? Or just one single measurement taken? More info is required.

P7 L 39: members

P10 L line 45: non specialist volunteers to estimate size of *C. carcharias* should be used

Author's Response to Decision Letter for (RSOS-190456.R0)

See Appendix A.

Decision letter (RSOS-190456.R1)

26-Apr-2019

Dear Mr May,

I am pleased to inform you that your manuscript entitled "Eyes on the size: Accuracy of visual length estimates of white sharks, *Carcharodon carcharias*" is now accepted for publication in Royal Society Open Science.

Kind regards,

Andrew Dunn

on behalf of Prof Kevin Padian (Subject Editor)

Appendix A

Response to reviewer's comments

Reviewer 1

Interesting and needed paper. Good job! References Section needs some cleaning up. Some incomplete, at least one has date listed at end instead of after authors.

Response: We thank the reviewer for their time reviewing the manuscript and complimentary comments about our manuscript. The reference list has been corrected and completed.

Reviewer 2

1) P5 – L22-55. Regarding accurate measuring of white sharks, it has been previously demonstrated that using stereo-video is a suitable and accurate method however there is no mention of this. There is no mention of other studies that have employed non-visual methods for estimating white shark size, even though there are published studies that document direct measurements rather than visual, there should be at least a sentence in the Intro discussing these other studies of direct measurement. See for example: Harasti, D., Lee, KA., Laird, R., Bradford, R. and Bruce., B. (2016). Use of baited remote underwater video systems (BRUVs) to estimate relative abundance and size of juvenile white sharks *Carcharodon carcharias*. Marine and Freshwater Research. <http://dx.doi.org/10.1071/MF16184>.

Response: A paragraph discussing other studies that have employed non-visual methods for estimating white shark size has been added in which the suggested reference has been added: “Stereo-photogrammetry has recently been used to measure a broad range of shark species including *C. carcharias* [43], *R. typus* [14], oceanic whitetip sharks, *Carcharhinus longimanus* [8] and sandbar sharks, *Carcharhinus plumbeus* [44], as it replaces the need to capture individuals. Stereo-photogrammetry uses two cameras recording simultaneous footage and specialised software to record point-to-point measurements [45]. The method is effective underwater, with Harvey *et al.* [46] quantifying the accuracy of the technique by comparing stereo-obtained measurements of southern bluefin tuna, *Thunnus maccoyii*, to true length measured directly, and found a difference of 0.16%. Despite the proven ability of stereo-camera technology to capture accurate measurements of *C. carcharias*, visual estimates are still favoured as a method due to the ease and reduced cost.”

We thank the reviewer for their comment and their efforts finding a detailed source to contribute to the discussion on stereo-camera technology.

2) P6 37-55: There needs to be much more information provide about the stereo-camera system used in regards how were they calibrated, what were the calibration tests and what was the camera accuracy. Below are some specific questions

How were the cameras calibrated? Was a calibration cube used?

How often was the camera calibration checked? Was it done just once or throughout the entire study? Did the camera need to recalibrated at all?

Was a known length scale bar used to test calibration?

How do you know that the cameras were providing accurate estimates?

Provide error estimates for the calibration tests and provide some data summarising the calibration tests. This will help demonstrate the accuracy of measurements from the stereo-camera system.

Can you provide an estimate of the size of the field of view or detection range of the

cameras?

Response: The following text was added to the methods in response to the reviewer's comments:

“The stereo-camera was calibrated with the CAL-Stereo Camera Calibration software, a distance bar, and calibration cube (1000 x 1000 x 500 mm; www.seagis.com.au). With the stereo-camera immobilised underwater, the calibration cube was manually rotated in different positions within the field of view of both cameras to ensure a range of calibration points at varying angles and distances. The distance bar is a horizontal bar 1.3 m long that is used to independently check calibration accuracy by constantly rotating and shifting the angle of the bar while moving closer to the camera. Recalibration of the stereo-camera was undertaken three times during the study period, with approximately six months between each calibration, to avoid inaccurate measurements due to shifts in camera angles from handling and operating the stereo-camera units. The mean level of error between true length and stereo-camera measurement was 0.42 cm, indicating a very high level of accuracy and reliability.”

How many measurements of each shark were used? Was an average measurement taken based on different passes/angle/frames? Or just one single measurement taken? More info is required.

The following text was added to address the reviewer's questions: “Each individual shark was measured in as many suitable passes as possible during each video, which resulted in a different number of measurements taken for each shark. Average measurement of all passes was calculated for each shark.”

3) P7 L 39: members

Response: Corrected to “members”

4) P10 L line 45: non specialist volunteers to estimate size of *C. carcharias* should be used

Response: Corrected to “non-specialist volunteers to estimate size of *C. carcharias* should be used”.